# Association between *Toxoplasma gondii* Exposure and Suicidal Behavior in Patients Attending Primary Health Care Clinics

**DOI:** 10.3390/pathogens10060677

**Published:** 2021-05-30

**Authors:** Cosme Alvarado-Esquivel, Sergio Estrada-Martínez, Agar Ramos-Nevárez, Alma Rosa Pérez-Álamos, Isabel Beristain-García, Ángel Osvaldo Alvarado-Félix, Sandra Margarita Cerrillo-Soto, Antonio Sifuentes-Álvarez, Gustavo Alexis Alvarado-Félix, Carlos Alberto Guido-Arreola, Leandro Saenz-Soto

**Affiliations:** 1Biomedical Research Laboratory, Faculty of Medicine and Nutrition, Juárez University of Durango State, Avenida Universidad S/N, Durango 34000, Mexico; aoafangel@hotmail.com (A.O.A.-F.); sifual55@hotmail.com (A.S.-A.); gaafalexis15@gmail.com (G.A.A.-F.); 2Institute for Scientific Research “Dr Roberto Rivera-Damm”, Juárez University of Durango State, Avenida Universidad S/N, Durango 34000, Mexico; semdurango@hotmail.com (S.E.-M.); almaross1@yahoo.es (A.R.P.-A.); 3Clínica de Medicina Familiar, Instituto de Seguridad y Servicios Sociales de los Trabajadores del Estado, Predio Canoas S/N, Durango 34079, Mexico; agar_ramos@hotmail.com (A.R.-N.); agar.ramos@issste.gob.mx (S.M.C.-S.); cguido@issste.gob.mx (C.A.G.-A.); quim_saenz@hotmail.com (L.S.-S.); 4Facultad de Enfermería y Obstetricia Juárez University of Durango State, Blvd, Juan Pablo II 512, Durango 34000, Mexico; beristaingarcia@yahoo.com.mx

**Keywords:** suicidal behavior, cross-sectional study, primary care, seroprevalence, epidemiology

## Abstract

This study aimed to determine the association between suicidal behavior and *T. gondii* seroreactivity in 2045 patients attending primary care clinics. IgG antibodies against *T. gondii* were found in 37 (12.1%) out of 306 individuals with a history of suicidal ideation and in 134 (7.7%) of 1739 individuals without this history (OR: 1.64; 95% CI: 1.11–2.42; *p* = 0.01). Seropositivity to *T. gondii* was associated with suicidal ideation in women (OR: 1.56; 95% CI: 1.01–2.42; *p* = 0.03) and individuals aged ≤30 years (OR: 3.25; 95% CI: 1.53–6.88; *p* = 0.001). No association between the rates of high (>150 IU/mL) levels of anti-*T. gondii* IgG antibodies and suicidal ideation or suicide attempts was found. IgG antibodies against *T. gondii* were found in 22 of 185 (11.9%) individuals with a history of suicide attempts and in 149 (8.0%) of 1860 individuals without this history (OR: 1.54; 95% CI: 0.96–2.49; *p* = 0.06). The seroprevalence of *T. gondii* infection was associated with suicide attempts in individuals aged 31–50 years (OR: 2.01; 95% CI: 1.09–3.71; *p* = 0.02), and with more than three suicide attempts (OR: 4.02; 95% CI: 1.34–12.03; *p* = 0.008). Our results indicate that *T. gondii* exposure is associated with suicidal behavior among patients attending primary care clinics.

## 1. Introduction

The coccidian protozoan *Toxoplasma gondii* (*T. gondii*) infects more than 100 species of vertebrates, including one-third of the human population [1,2]. Toxoplasmosis, the disease that is caused by *T. gondii*, is a zoonotic disease of global distribution and importance [3]. Cats are the most important host in the epidemiology of toxoplasmosis because they are the only species that can excrete oocysts in feces [4]. The main routes of transmission are by ingestion of tissue cysts in raw or undercooked meat of infected animals, ingestion of raw vegetables or water contaminated with *T. gondii* oocysts from cat feces, and transplacental [3]. Infection with *T. gondii* in an immunocompetent host does not typically show symptoms, and parasites are retained in latent tissue cysts that can be reactivated upon immune suppression and could damage key organ systems [2]. Some patients with toxoplasmosis present cervical lymphadenopathy or ocular disease [5]. Toxoplasmosis can be fatal to the fetus and immunocompromised adults [3]. The reactivation of latent disease in immunocompromised patients can cause life-threatening encephalitis [5]. In addition, toxoplasmosis has been linked to a range of behavioral alterations and conditions [1]. *T. gondii* has a preference for invading neurons and affecting the functioning of glial cells [6]. The seropositivity to *T. gondii* has been associated with mixed anxiety and depressive disorder [7], schizophrenia [8,9,10], obsessive-compulsive disorder [11,12], and an increased risk of traffic accidents [13]. In a study of decedents in Poland, the researchers found a strong correlation between latent *T. gondii* infection and engaging in risky behaviors leading to death [14]. Furthermore, suicide behavior in psychiatric patients has been associated with high titers of anti-*T. gondii* antibodies [15,16] and the seroprevalence of *T. gondii* infection [17]. To the best of our knowledge, the link between *T. gondii* infection and suicide behavior in patients of primary care has not been studied. The aim of this study was to determine the association between suicidal behavior and *T. gondii* infection in outpatients that were attending primary health care clinics in Durango, Mexico.

## 2. Results

Out of the 2045 individuals studied, 306 (15.0%) had a history of suicidal ideation and 1739 (85.0%) did not have this history. IgG antibodies against *T. gondii* were found in 37 (12.1%) of the 306 individuals with a history of suicidal ideation and in 134 (7.7%) of the 1739 individuals without this history (OR: 1.64; 95% CI: 1.11–2.42; *p* = 0.01). Table 1 shows a stratification by age and sex and seroprevalence of *T. gondii* infection in individuals with and without a history of suicidal ideation. Women with a history of suicidal ideation had a significantly higher (29/251: 11.6%) seroprevalence of *T. gondii* infection than women without this history (111/1445: 7.7%) (OR: 1.56; 95% CI: 1.01–2.42; *p* = 0.03). Individuals that were aged ≤ 30 years with a history of suicidal ideation had a significantly higher seroprevalence of *T. gondii* infection than those of the same age group without suicidal ideation (13/87: 14.9% vs 19/371: 5.1%, respectively) (OR: 3.25; 95% CI: 1.53–6.88; *p* = 0.001).

High (>150 IU/mL) levels of anti-*T. gondii* IgG antibodies were found in 15 (4.9%) of the 306 individuals with a history of suicidal ideation and in 50 (2.9%) of the 1739 individuals without this history (OR: 1.74; 95% CI: 0.96–3.14; *p* = 0.06). Table 2 shows a stratification by sex and age groups and the association between high (>150 IU/mL) anti-*T. gondii* IgG antibody levels and suicidal ideation. A borderline association between high levels of anti-*T. gondii* IgG antibodies and suicidal ideation in individuals that were aged ≤30 years was found (OR: 3.17; 95% CI: 0.98–10.24; *p* = 0.05).

Anti-*T. gondii* IgM antibodies were found in 10 (27.0%) of the 37 individuals with anti-*T. gondii* IgG antibodies and a history of suicidal ideation and in 26 (19.4%) of the 134 individuals with anti-*T. gondii* antibodies without this history (OR: 1.53; 95% CI: 0.66–3.57; *p* = 0.31).

With respect to suicide attempts, of the 2045 individuals that were studied, 185 (9.0%) had a history of suicide attempts and 1860 (91.0%) did not have this history. IgG antibodies against *T. gondii* were found in 22 of the 185 (11.9%) individuals with a history of suicide attempts and in 149 (8.0%) of the 1860 individuals without this history (OR: 1.54; 95% CI: 0.96–2.49; *p* = 0.06). Table 3 shows a stratification by age and sex and the seroprevalence of *T. gondii* infection in individuals with and without a history of suicide attempts. Stratification by sex showed no association between *T. gondii* infection and suicide attempts. Whereas stratification by age groups showed that individuals aged 31–50 years with a history of suicide attempts had a significantly higher (14/89: 15.7%) seroprevalence of *T. gondii* infection than individuals of the same age group without a history of suicide attempts (89/1050: 8.5%) (OR: 2.01; 95% CI: 1.09–3.71; *p* = 0.02).

High (>150 IU/mL) levels of anti-*T. gondii* IgG antibodies were found in nine (4.9%) of the 185 individuals with a history of suicide attempts and 56 (3.0%) of the 1860 individuals without this history (OR: 1.64; 95% CI: 0.80–3.38; *p* = 0.17). Table 4 shows a stratification by sex and age groups and the association between high (>150 IU/mL) anti-*T. gondii* IgG antibody levels and a history of suicide attempts. No association between the rates of high levels of anti-*T. gondii* antibodies and a history of suicide attempts among sex and age groups was found.

Anti-*T. gondii* IgM antibodies were found in eight (36.4%) of the 22 individuals with anti-*T. gondii* IgG antibodies and a history of suicide attempts and in 28 (18.8%) of the 149 individuals with anti-*T. gondii* antibodies without a history of suicide attempts (OR: 2.46; 95% CI: 0.94–6.45; *p* = 0.05). Table 5 shows a stratification by age and sex and seropositivity to anti-*T. gondii* IgM antibodies in the participants with anti-*T. gondii* IgG antibodies with and without a history of suicide attempts.

Individuals with a history of suicide attempts have attempted suicide from one to 20 (median 1) times. Individuals with more than three suicide attempts had a significantly higher seroprevalence of *T. gondii* infection than those with 1–3 suicide attempts (6/20: 30.0% vs 15/156: 9.6%, respectively) (OR: 4.02; 95% CI: 1.34–12.03; *p* = 0.008). The frequency of high anti-*T. gondii* IgG antibodies in individuals with more than three suicide attempts was similar to that found in individuals with 1–3 suicide attempts (2/20: 10.0% vs 6/156: 3.8%, respectively) (OR: 2.77; 95% CI: 0.52–14.80; *p* = 0.22). The date (years before the interview) of the last suicide attempt was not associated with the seroprevalence of *T. gondii* infection (*p* = 0.76), high levels of anti-*T. gondii* IgG antibodies (*p* = 1.00), or anti-*T. gondii* IgM antibodies (*p* = 0.21). The methods of suicide attempts (wounding, hanging, drug overdose, poisoning, firearm, electrocution, fasting, or throwing themselves to vehicles) were not associated with the seroprevalence of *T. gondii* infection (*p* = 0.53), high levels of anti-*T. gondii* IgG antibodies (*p* = 0.50), or anti-*T. gondii* IgM antibodies (*p* = 0.33).

## 3. Discussion

Most of the people (91.7%) who commit suicide had a health care contact with an emergency room visit, primary care, or outpatient specialty setting within a year prior to suicide [18]. Thus, it is important to study suicide behavior in these settings. In this study, we sought to determine the association between suicide behavior and *T. gondii* exposure in outpatients attending primary health care clinics. We found that the seroprevalence of *T. gondii* infection was significantly higher in individuals with a history of suicidal ideation than in those without this history. Therefore, the results suggest that infection with *T. gondii* is associated with suicidal ideation in patients attending primary health care clinics. Our results agree with those that were reported in a study of children and adolescences with depression, where researchers found that seropositivity to *T. gondii* was significantly higher in patients with suicidal ideation than in those without suicidal ideation [19]. In contrast, our results conflict with those that were reported in some studies. A negative association between seroprevalence of *T. gondii* infection and suicidal ideation was found in patients suffering from mental and behavioral disorders due to psychoactive substance use [20]. The difference in the association among the studies could be due to the difference in the characteristics of the populations studied. In the study of patients with mental and behavioral disorders, patients were treated, and this condition might have prevented suicidal ideation. In a National Health and Nutrition Survey of 5 487 subjects that were aged 20 to 80 years in the USA, neither *T. gondii* seroprevalence nor anti-*T. gondii* antibody titer was positively associated with suicidal ideation [21]. The difference in the characteristics of the populations studied might explain the difference in the association among the studies. For instance, our study population consisted of outpatients attending primary care, whereas in the American study, people from the general population of the U.S. were studied. Regarding suicide attempts, no association between *T. gondii* infection and suicide attempts in general was found. However, stratification by age groups showed that *T. gondii* infection was associated with suicide attempts in individuals that were aged 31–50 years. No association between *T. gondii* seropositivity and suicide attempts was found in other studies, including psychiatric outpatients [16] and patients with recurrent mood disorders [15]. In contrast, a positive association between *T. gondii* seropositivity and suicide attempts has been found in women of postmenopausal age [22] and psychiatric patients [17]. In two recent meta-analyses, researchers confirmed that infection with *T. gondii* is a potential risk factor for suicidal behavior [23,24]. We did not find an association between high levels of anti-*T. gondii* IgG antibodies and suicidal ideation or suicide attempts in general. However, a borderline (*p* = 0.05) association between high antibody levels and suicidal ideation in individuals that were aged ≤30 years was found. High levels of anti-*T. gondii* IgG antibodies have been associated with suicide attempts in patients with recurrent mood disorders [15], psychiatric outpatients [16], and patients with schizophrenia younger than 38 years [25]. Remarkably, we found an association between seropositivity to *T. gondii* and a high (>3) number of suicide attempts. This finding agrees with the one that was found in a study of psychiatric outpatients, where the rate of *T. gondii* seropositivity increased with the number of suicide attempts [16]. In contrast, in a study of patients with recurrent mood disorders, no significant relationship was found between *T. gondii* seropositivity and number of prior suicide attempts [15]. In the present study, we found a borderline (*p* = 0.05) association between seropositivity to anti-*T. gondii* IgM antibodies in the participants with anti-*T. gondii* IgG antibodies and suicide attempts. This finding suggests that a recent *T. gondii* infection could be associated with suicide attempts. If our findings are confirmed by further studies, then physicians providing primary care consultations might consider testing for *T. gondii* exposure of their patients with suicidal behavior and those with suicide risk factors, including, for instance, people with depression, anxiety, or with a family history of suicide attempts.

## 4. Materials and Methods

Through a cross-sectional study design, we surveyed 2045 people attending primary health care clinics in Durango, Mexico, from 2014 to 2019. The inclusion criteria for enrollment in the survey were: (1) people attending primary health care clinics in Durango, Mexico; (2) any gender; (3) aged ≥ 15 years; and, (4) with a written informed consent. Of the 2045 people that were enrolled in the survey, 1696 were women and 349 were men. The mean age of participants was 40.29 ± 12.78 (range 15–84) years.

In-person interviews were conducted for the survey data collection. Participants were asked about suicidal behavior, including a history of suicidal ideation, suicide attempts, number of suicide attempts, date of last suicide attempt, and the methods used for suicide attempt.

The participants provided a blood sample that was centrifuged, and the sera obtained was frozen at −20 ℃ until analyzed. All of the serum samples were analyzed for the detection of anti-*T. gondii* IgG antibodies using a commercially available enzyme immunoassay “*Toxoplasma gondii* IgG” kit (Diagnostic Automation/Cortez Diagnostics, Inc., Woodland Hills, CA. USA). Serum samples with the presence of anti-*T. gondii* IgG antibodies were further analyzed for the detection of anti-*T. gondii* IgM antibodies by a commercially available enzyme immunoassay “*Toxoplasma gondii* IgM” kit (Diagnostic Automation/Cortez Diagnostics, Inc.). All of the tests were performed following the instructions of the manufacturer. Positive and negative controls that were provided in the kits were included in each run. The serum samples were analyzed soon (no more than two months) after freezing.

Analysis of data was performed with the software IBM SPSS Statistics version 20 and Epi Info version 7. Sample size was calculated using the following values: a population size of 300000, an expected frequency of *T. gondii* exposure of 6.1% [26], confidence limits of 2%, a design effect of 1, and a confidence level of 99.9%. A sample size of 1542 was obtained. The Fisher’s exact test (for small values) and Pearson’s chi-square test were used to compare the frequencies of seropositivity and serointensity among the groups. The association between variables was assessed by calculating the odd ratios (OR) and 95% confidence intervals (CI), and a *p* value of less than 0.05 was considered to be statistically significant.

## 5. Conclusions

Our results indicate that *T. gondii* exposure is associated with suicidal behavior among people attending primary care clinics. Our findings further support the association between *T. gondii* infection and suicidal behavior that was reported in other population groups.

## Figures and Tables

**Table 1 pathogens-10-00677-t001:** Association between *T. gondii* exposure and suicidal ideation, a stratification by sex and age groups.

.	Suicidal Ideation			No Suicidal Ideation					
		Seropositivity			Seropositivity			95%	
	No.	to *T. gondii*		No.	to *T. gondii*			Confidence	*p*
Characteristic	Tested	No.	%	Tested	No.	%	OR	Interval	Value
Sex									
Male	55	8	14.5	294	23	7.8	2.00	0.84–4.74	0.10
Female	251	29	11.6	1445	111	7.7	1.56	1.01–2.42	**0.03**
Age (years)									
≤30	87	13	14.9	371	19	5.1	3.25	1.53–6.88	**0.001**
31–50	166	18	10.8	973	85	8.7	1.27	0.74–2.17	0.38
>50	53	6	11.3	395	30	7.6	1.55	0.61–3.92	0.34
All	306	37	12.1	1739	134	7.7	1.64	1.11–2.42	**0.01**

**Table 2 pathogens-10-00677-t002:** Association between high (>150 IU/mL) levels of anti-*T. gondii* IgG. antibodies and suicidal ideation, a stratification by sex and age groups.

	Suicidal Ideation			No Suicidal Ideation					
		>150 IU/mL			>150 IU/mL			95%	
	No.	of IgG		No.	of IgG			Confidence	*p*
Characteristic	Tested	No.	%	Tested	No.	%	OR	Interval	Value
Sex									
Male	55	2	3.6	294	6	2.0	1.81	0.35–9.21	0.61
Female	251	13	5.2	1445	44	3.0	1.73	0.92–3.27	0.08
Age (years)									
≤30	87	5	5.7	371	7	1.9	3.17	0.98–10.24	0.05
31–50	166	6	3.6	973	33	3.4	1.06	0.44–2.59	0.88
>50	53	4	7.5	395	10	2.5	3.14	0.94–10.40	0.07
All	306	15	4.9	1739	50	2.9	1.74	0.96–3.14	0.06

**Table 3 pathogens-10-00677-t003:** The association between T. gondii exposure and suicide attempts, a stratification by sex and age groups.

	Suicide Attempts			No Suicide Attempts					
		Seropositivity			Seropositivity			95%	
	No.	to *T. gondii*		No.	to *T. gondii*			Confidence	*p*
Characteristic	Tested	No.	%	Tested	No.	%	OR	Interval	Value
Sex									
Male	36	5	13.9	313	26	8.3	1.78	0.63–4.96	0.34
Female	149	17	11.4	1547	123	8.0	1.49	0.87–2.55	0.14
Age (years)									
≤30	64	6	9.4	394	26	6.6	1.46	0.57–3.71	0.41
31–50	89	14	15.7	1050	89	8.5	2.01	1.09–3.71	**0.02**
>50	32	2	6.2	416	34	8.2	0.74	0.17–3.26	1.00
All	185	22	11.9	1860	149	8.0	1.54	0.96–2.49	0.06

**Table 4 pathogens-10-00677-t004:** Association between high (>150 IU/mL) levels of anti-*T. gondii* IgG antibodies and suicide attempts, a stratification by sex and age groups.

	Suicide Attempts			No Suicide Attempts					
		>150 IU/mL			>150 IU/mL			95%	
	No.	of IgG		No.	of IgG			Confidence	*p*
Characteristic	Tested	No.	%	Tested	No.	%	OR	Interval	Value
Sex									
Male	36	1	2.8	313	7	2.2	1.24	0.14–10.45	0.58
Female	149	8	5.4	1547	49	3.2	1.73	0.80–3.73	0.15
Age (years)									
≤30	64	3	4.7	394	9	2.3	2.10	0.55–7.98	0.22
31–50	89	5	5.6	1050	34	3.2	1.77	0.67–4.66	0.22
>50	32	1	3.1	416	13	3.1	1.00	0.12–7.89	1.00
All	185	9	4.9	1860	56	3.0	1.64	0.80–3.38	0.17

**Table 5 pathogens-10-00677-t005:** Association between *T. gondii* IgM seropositivity in participants with anti- *T. gondii* IgG antibodies and suicide attempts, a stratification by sex and age groups.

	Suicide Attempts			No Suicide Attempts					
		Seropositivity			Seropositivity			95%	
	No.	to *T. gondii* IgM		No.	to *T. gondii* IgM			Confidence	*p*
Characteristic	Tested	No.	%	Tested	No.	%	OR	Interval	Value
Sex									
Male	5	2	40.0	26	2	7.7	8.00	0.80–79.65	0.11
Female	17	6	35.3	123	26	21.1	2.03	0.68–6.02	0.19
Age (years)									
≤30	6	2	33.3	26	8	30.8	1.12	0.16–7.45	1.00
31–50	14	5	35.7	89	18	20.2	2.19	0.65–7.34	0.29
>50	2	1	50.0	34	2	5.9	16.00	0.70–361.7	0.16
All	22	8	36.4	149	28	18.8	2.46	0.94–6.45	0.05

## Data Availability

Data is provided within the article.

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
