# Peer review of "Association between Toxoplasma gondii Exposure and Suicidal Behavior in Patients Attending Primary Health Care Clinics"

_pathogens, 2021, doi:10.3390/pathogens10060677_

Round 1
Reviewer 1 Report
Overview of Manuscript: In this study, Alvarado-Esquivel et al., aimed to determine if there is an association between suicidal behavior and T. gondii seropositivity in patients attending primary care clinics. This work was an extension of several prior studies showing a correlation between high titers of anti-T. gondii antibodies and seropositivity of T. gondii infection and suicide. This study was conducted in Durango, Mexico and done with a cross-sectional study design stratified by age and sex, consisting of 2045 patients, with 1696 females and 349 males. Survey data was used collect data on suicidal behavior. Blood samples were collected and anti-T. gondii IgG and IgM antibodies detected by ELISA. The data presented in 4 Tables using an Odds Ratio method to determine significance of results.
The major findings of this study were: 1). An association was found between individuals seropositive for T. gondii and suicidal ideation in primary health care clinics in women, individuals aged less than 30 yrs, 2). No association was found between high anti-T. gondii IgG titers and suicidal ideation or suicide attempts and 3). An association was found between T. gondii seropositivity and history of suicide attempts in those aged 31-50 yrs of age. The general conclusion of this study finds an association between T. gondii infection and suicidal behavior, thus supporting work from previous studies. This study is the first to show an association of suicidal behavior and T. gondii infections in outpatients attending primary health care clinics in Durango Mexico. This result is of importance as most people who commit suicide had a health care contact with an emergency room visit, primary care or outpatient setting. While some of the findings support the previous studies investigating an association between T. gondii seropositivity and suicide ideation, some results conflicted with previous studies, most notably a larger study (n= 5487) done in the US. These different results might reflect differences in the characteristics of the populations studied, which is of potential significance is assessing T. gondii seropositivity and suicide risk. This study also contributed some novel findings such as the correlation between T. gondii seropositivity and suicide attempts in the 31-50 yr. age group.
This study is well done with appropriate statistical analysis, with the data well-presented and discussed in the manuscript, with only a few minor caveats. The manuscript contributes some novel findings and some confirmation of previous studies. The results and data presented in this manuscript will be of interest to the scientific community and specifically those working in health care settings dealing with mental illnesses.
Minor Issues to be addressed:
- Results section, p. 2, line 62: the sentence beginning ‘1739 (%)’ a number should be included before the ‘%’.
- Data section p. 5; lines 120-124: The results on IgM data is mentioned in the text but the data is not included in any of the tables or further discussed in the Discussion section. What is the authors interpretation of the significance of IgM antibodies present in those with IgG titers and a history of suicidal attempts (OR = 2.46 and p value = 0.05)?
Author Response
- Results section, p. 2, line 62: the sentence beginning ‘1739 (%)’ a number should be included before the ‘%’.
A number (85.0) before the % was added (line 62).
- Data section p. 5; lines 120-124: The results on IgM data is mentioned in the text but the data is not included in any of the tables or further discussed in the Discussion section. What is the authors interpretation of the significance of IgM antibodies present in those with IgG titers and a history of suicidal attempts (OR = 2.46 and p value = 0.05)?
A new Table (6) with IgM data was added (lines 124-129).
An interpretation of the significance of IgM antibodies present in those with IgG antibodies and a history of suicide attempts was added to the Discussion section (191-194).
Thank you for your valuable comments for improving our manuscript.

Reviewer 2 Report
This is a valuable paper about the relationship between T. gondii infection and suicide risk. The authors were done a great job involved such a large group. The study is really interesting, and the results have high significance of content
My suggestions:
Please, mark the significance of the results in the table – it makes them more visible.
Were all serum samples analyzed at this same time point? Please add this information to the methods section. Storage samples in -20 for five years (from 2014 to 2019) could affect results. Please add information about the maximum time of storage samples in these conditions.
To my mind, in conclusion, or discussion, there is missing in the practical or clinical translation of obtained results – performing more often screening T. gondii IgG f.e. in risk population (with depression, family history of suicide attempts?)
Author Response
- Please, mark the significance of the results in the table – it makes them more visible.
Significant P values were marked with bold numbers in Tables (lines 72-73, 107-108).
- Were all serum samples analyzed at this same time point? Please add this information to the methods section. Storage samples in -20 for five years (from 2014 to 2019) could affect results. Please add information about the maximum time of storage samples in these conditions.
Serum samples were analyzed soon (no more than 2 months) after freezing (lines 217-218).
- To my mind, in conclusion, or discussion, there is missing in the practical or clinical translation of obtained results – performing more often screening T. gondii IgG f.e. in risk population (with depression, family history of suicide attempts?)
A clinical translation of the obtained results was added to the Discussion section (lines 194-198).
Thank you for your valuable comments for improving our manuscript.
